# DUAL-LEVEL PROTOTYPES GUIDANCE FOR SINGLE-FRAME TEMPORAL ACTION LOCALIZATION

## ABSTRACT

In recent, single-frame temporal action localization (STAL) has captured the attention of the computer vision community. Due to the sparse single-frame annotations, current STAL methods generally employ pseudo-labels strategies to bridge the gap between weakly-supervised methods and fully-supervised methods. However, these methods derive pseudo-labels from single-frame of the corresponding instances, yet the intra-class affinity from the current single-frame to other action snippets remains neglected. To capitalize on this affinity, we design a dual-level prototypes guidance (DPG) method with the graph matching random walk (Gm-Rw) algorithm to achieve instance-level and video-level prototype guidance for pseudo-labels refinement. For instance-level guidance, the Gm-Rw exploits the high affinity prototype among instances of the current video to build intra-class associations. For video-level guidance, an online memory bank is constructed to iteratively summarize more discriminative prototype. After Gm-Rw builds affinity among intra-class videos, an exponential moving average (EMA) mechanism is designed to achieve dual-level prototypes guidance for pseudo-labels refinement. Notably, the dual-level guidance is mutually reinforcing, prompting us to propose a novel adaptive collaborative strategy (ACS) for dynamic optimization. Extensive experiments on THUMOS14, GTEA, BEOID, and ActivityNet1.3 reveal that our method significantly outperforms state-of-the-art methods.

## 1 INTRODUCTION

Temporal action localization (TAL) Field et al. (2007); Ma et al. (2005); Vishwakarma & Agrawal (2013) is one of the most fundamental tasks of video understanding, which aims to localize start and end timestamps of actions within video sequences. Thanks to precise boundary annotations, the fully-supervised TAL (FTAL) Dai et al. (2017); Long et al.; Shou et al. (2017); Chao et al. (2018); Zhao et al. (2020); Sridhar et al. (2021) methods have triggered remarkable progress. However, the frame-level annotation of action instances is labor-intensive and time-consuming. To mitigate this issue, learning with only video-level labels, weakly-supervised temporal action localization (WTAL) Zhang et al. (2020); Narayan et al. (2020); Luo et al. (2021); Ma et al. (2021); Huang et al. (2021); Luo et al. (2020); Li et al. (2022a) has drawn considerable interest recently. Conventionally, WTAL follows a localization and classification pipeline based on snippet-wise classification score but coarse-grained labels limit its performance. Fortunately, the single-frame temporal action localization (STAL) Lee & Byun (2021); Fu et al. (2022); Yu et al. (2023); Li et al. (2024) bridges a gap between FTAL and WTAL tasks, adding only a timestamp for each video instance under WTAL paradigm. This pioneering research consumes almost comparable labor costs as video-level annotations while providing more pronounced localization results. This prompted us to address TAL from the perspective of STAL. **We provide more related work about TAL in appendix A.1.**

Special single-frame annotation for STAL provides sparse location information but brings important feature representation. In this case, pseudo-labels methods are proposed, which exploit pseudo action snippets from the WTAL paradigm to perform FTAL optimization. These pseudo-labels methods utilize similarity metrics to derive pseudo-labels based on single-frame features, which are then propagated using dynamic programming algorithms. Unfortunately, this sparse annotation causes sub-optimal single-frame feature representation, the following issues naturally arise: ❶ Insufficient feature discrimination leads to an inferior metrics process; and ❷ Lacking feature completeness hinders dynamic propagation. Tackling these two issues could provide insights into the compre-

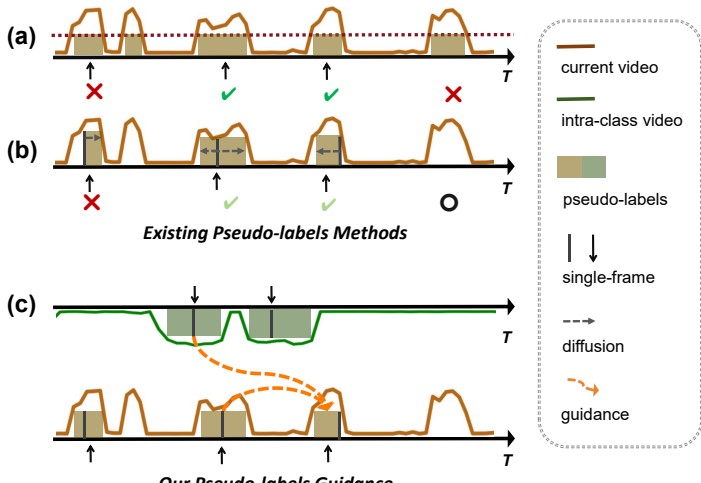

Figure 1: Comparison of pseudo-labels methods. (a) illustrates the pseudo-labels method in STAL based on threshold segmentation and single-frame filtering methods. (b) shows the STAL methods for generating pseudo-labels based on similarity metrics for a single frame of the corresponding instance. (c) briefly shows the way we proceeded for the pseudo-labels guidance, where the intra-class affinity prototype is utilized.

hensive use of single-frame, and motivate us to improve the pseudo-labels from a prototype view. More directly, single-frame features are regarded as non-learnable prototypes. To solve issue ❶, the single-frame features within the current video are regarded as instance-level prototype, which have high affinity among instances and can improve discrimination. As for issue ❷, we construct a memory bank to collect and sort high discriminative single-frame features from intra-class videos as video-level prototype, and diversity supports the feature completeness. After dual-level prototypes summarizing, i.e. instance-level and video-level, we design a novel graph matching random walk (Gm-Rw) algorithm to match and enhance the representations of action snippets continuously. Gm-Rw algorithm mines node-to-node correspondences in graph structures based on the inherent similarity details and then gets a higher quality single frame based on instance-level and video-level prototype guidance. Importantly, the instance-level prototype leads to stable and sustainable representation enhancement. Differently, the video-level prototype depends on the memory bank's storage, which is a progressive process when updating discrimination. Thus, dual-level guidance is mutually complementary, and we additionally propose an adaptive collaborative strategy that dynamically optimizes our dual-level prototypes guidance.

The prototype guidance has some appealing qualities. **Stability**: The sources of the prototype are directly accessible, which can support a non-learnable prototype and ensure the reliability of features. The kinds of prototype are diverse (e.g., instance-level prototype, video-level prototype), which can prompt the model to learn more action features and ensure the effectiveness of the experiments. **Explainability**: The proposed dual-level prototypes guidance falls under the category of non-learnable prototype and holds natural explainability by continuous guidance through similarity-based metrics. Therefore, the proposed model can make explainable inferences, which is ahead of most models that failed to expound exactly the model's working process. **Flexibility**: Dual-level prototypes guidance owns a complementary collaborative strategy that can meet the requirements of the weight of different tasks. Its core epistemology is learning from data at different granularity (e.g., instance-level learning, video-level learning, etc.) to achieve a generalized vision framework.

In summary, our contributions are three-fold:

(1) We rethink to refine pseudo-labels for STAL in a prototype view using the Gm-Rw algorithm. For instance-level guidance, we leverage the affinity between single-frame features and action instances to enhance action representations. For video-level guidance, we summarize and store more discriminative prototypes to provide intra-class prototype guidance.

(2) The DPG is collaborative, thus we additionally design an adaptive collaborative strategy (ACS) to achieve more comprehensive prototype guidance.

(3) Comprehensive experiments on four standard benchmarks demonstrate that our proposed method achieves state-of-the-art results.

## 2 EXISTING PSEUDO-LABELS METHODS FOR STAL

**Point-level pseudo-labels.** Point-level pseudo-labels of STAL refers to using a single-frame label as the action label in each action instance while following the WTAL paradigm for training. In the point-level weakly-supervision setting, each action instance is annotated with only a timestamp, showing negligible additional cost compared to video-level instances. Ma et al. (2020); Lee & Byun (2021) starts with single-frame annotation, mines pseudo action frames and background frames, and constructs point-level classification results, making full use of single-frame supervision to train the model. Experimental results show that compared with full-supervision annotation, single-frame annotation greatly saves annotation time and can effectively improve the WTAL model. Its performance is even better than the full-supervision method.

**Instance-level pseudo-labels.** Although point-level supervision can provide more action-related information, due to the sparsity of single-frame annotations, the model in this mode still cannot obtain complete action instances. To solve the quality problem of single-frame annotations, Fu et al. (2022) starts from the similarity of single-frame and considers that the features of different fragments in action instances should be similar. We try to use point-level annotations to mine training samples for feature learning to ensure that two similar samples in different feature spaces are similar, which enhances the compactness of feature representation and reduces intra-action variations.

**Pseudo-labels improving.** At present, although research has made progress in improving the quality of single-frame, due to the gap between classification and positioning models, pseudo-labels based on CAS mining will inevitably introduce noise. To solve this problem, Li et al. (2024) proposed a guidance strategy based on semantic neighborhood. The core idea is that the similarity of adjacent segments is relatively high. This guidance strategy can help improve the quality of pseudo-labels and thus suppress pseudo-labels noise. Experiments show that this strategy can obtain higher quality pseudo-labels and achieve good results.

The above studies mainly improve the quality of single-frame from a single instance or video. Although these methods have achieved results, they still face the problem of low single-frame quality. To solve this problem, we designed a dual-level prototypes guidance architecture from the perspective of non-learning prototypes, which constructs the discriminability and integrity of single-frame at both the instance-level and the video-level to improve the quality of single-frame and has been effectively verified on the benchmarks THUMOS14, GTEA, and BEOID.

## 3 THE PROPOSED METHOD

In this section, we first introduce the problem formulation, as presented in section 3.1. Following this, a concise depiction of the preliminaries about the baseline network will be presented in section 3.2. Subsequently, our innovative dual-level prototypes guidance method will be provided in comprehensive detail in section 3.3. Lastly, the introduction of the adaptive complement strategy will be presented in section 3.4.

### 3.1 PROBLEM FORMULATION

We assume that a set of training videos is denoted as $\{v_i\}_{i=1}^{N}$, where $N$ represents the total number of training videos. Each video, $v_i$, is associated with a corresponding video label, $y_i \in \mathbb{R}^C$ and an additional timestamp, $\tau_i \in T$, where $C$ signifies the classes. During the testing phase, we endeavor to predict a set of action proposals, denoted as $\{(t_s, t_e, c, q)\}$ for each video. Here, $t_s$ and $t_e$ respectively denote the start and end times of an action instance, $c$ indicates the predicted action class, and $q$ represents the confidence score associated with the prediction.

## 3.2 PRELIMINARIES

**Baseline Setup.** We adopt extracted features as our input. The features include RGB features $F^{rgb} \in \mathbb{R}^{T \times D}$ and optical flow features $F^{flow} \in \mathbb{R}^{T \times D}$, where $T$ and $D$ are the number of snippets and the dimension, respectively. In addition, we obtain the corresponding single-frame features based on the single-frame annotations $\tau$. The video-level class label $p_i$ can be derived by accumulating the point-level labels using the temporal top-k pooling for aggregation. We only use $F$ to denote the two kinds of features in the following sections for simplicity and adopt LAC Lee & Byun (2021) as the baseline setting. **Appendix A.2 has more descriptions about the baseline.**

**Discussion.** Our selection of baseline adheres to the pseudo-labels method illustrated in Fig. 1. (b) of STAL, where pseudo-labels are generated based on the corresponding single frame features using a similarity metric. However, a problem arises with single-frame, as they represent specific frames within an action that may only capture one aspect of the action. When the single-frame label represents non-significant parts, the quality of the pseudo-labels is sub-optimal. The effectiveness of improving the model localization performance is heavily reliant on the quality of the pseudo-labels. Therefore, we must consider leveraging single-frame features from other action instances. The single-frame of instances can be derived from two sources: the instances from current video and other videos. This observation motivates us to address the issue at dual-level, i.e. instance-level and video-level . To this end, we propose a dual-level prototypes guidance strategy.

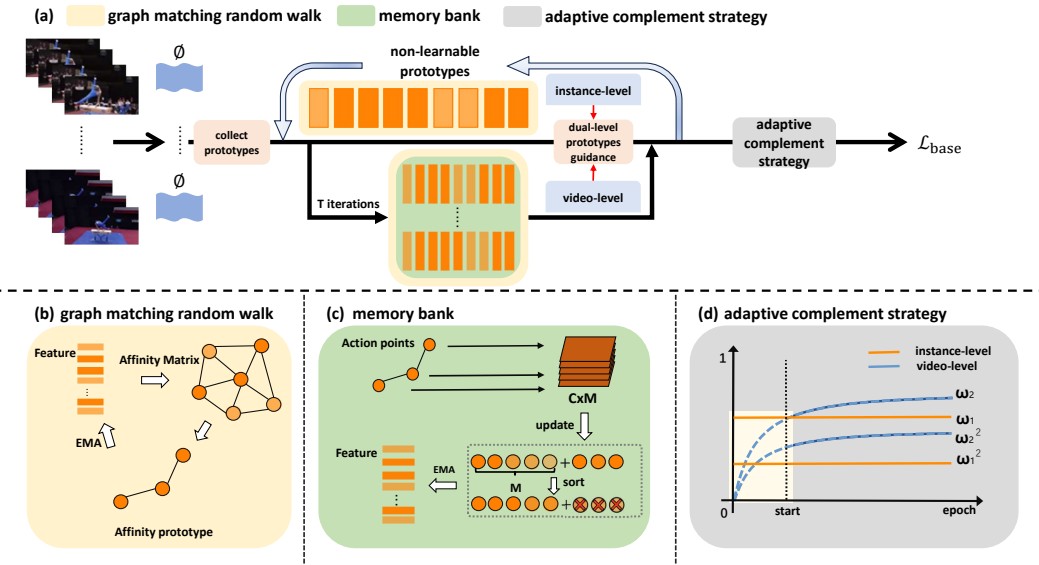

Figure 2: Network architecture. (a) The proposed network consists of the baseline Lee & Byun (2021) and two important modules, i.e. the dual-level prototypes guidance (DPG) and adaptive collaborative strategy (ACS). (b) The DPG uses the graph matching random walk algorithm (Gm-Rw) to realize affinity prototype guidance at the instance-level and video-level, respectively. (c) In video-level prototype guidance, a memory bank is proposed to store affinity prototypes of various videos. (d) ACS describes the weight changes for dual-level dynamic mutual optimum.

## 3.3 DUAL-LEVEL PROTOTYPES GUIDANCE

**Prototype summary.** Single-frame features exhibit highly salient action representations with strong activation in class activation sequences (CAS). Instances of the same action class demonstrate highly consistent representational patterns, leading to strong correlations from single-frame features to action snippets. These correlations are demonstrated to be both affine and highly discriminative for action recognition. we extract single-frame features for the action instances within the single-frame timestamps $\tau$ and form a frame sequence $F_{\tau}$. The current conceptual framework is constrained in its ability to assess more similar affinity solely within a single video. We leverage activation values from $CAS_{\tau}$ as an index for discriminative features, preserving intra-class prototype for each action

category. By exploiting the correspondence between $CAS_\tau$ and single-frame features, we establish two memory tables to store the single-frame features and their associated **scores**. Given the single-frame features of a video, we derive their prediction **scores** from action classification. The index of each class $C$ is denoted as:

$$Index_c = Sort(Scores \oplus CAS_\tau) \tag{1}$$

We reference the existing scores and their prediction scores of the corresponding single-frame features, retaining their index values. The $CAS_\tau$ serves as the foundation for updating and enhancing the **memory bank**, which is a structure to store the more discriminative affinity prototype and is updated throughout the Gm-Rw algorithm.

$$M_c = \operatorname{argmax}_{Index_c}(M_c \oplus F_\tau) \tag{2}$$

Where $\oplus$ means that we contact stored memory prototypes based on scores and single-frame features of current videos. Then we remain the most discriminative $M$ points for each class based on the index sorted values.

**Graph matching random walk (Gm-RW).** Currently, we will contemplate the comprehensive and efficient utilization of the summarized affinity prototype. In the field of computer vision, the Graph Matching (GM) Aldous & Fill (2002) algorithms are designed to identify node-to-node correspondences between graph structures by leveraging the similarity information inherent in these structures. Given that the GM seeks node-to-node matches, its outcomes are typically represented by an assignment matrix $X$. In the context of Single-frame Temporal Action Localization (STAL), individual single-frame timestamps $\tau$ are construed as nodes within the graph. The similarity matrix computed between single-frame timestamps and the current video can be viewed as a form of graph matching structure. This insight prompts us to explore the integration of the graph matching algorithms.

In our previous discussion, we summarized dual-level prototypes, which enable us to construct the affinity from single-frame features to action snippets as a node-to-node relationship for the current video. To establish this relationship, we try to build the assignment matrix for the GM algorithm which could realize instance-level and video-level prototype seeking and matching, as illustrated in Fig. 3. Thus, we derive two forms of node relation, namely the affinity association graph $A_a$ and the discriminative association graph $A_d$, representing the summarized prototype for the current video. The computation of these two matrices is outlined as follows:

$$A_a = Softmax(Norm_1(F_\tau * F^T))$$
$$A_d = Softmax(Norm_1(M_c * F^T)) \tag{3}$$

Here, we employ $Norm_1$ for normalization. By adhering to the principles of graph matching, there are two graphs with matching in the problem form of graph matching, whereas random walk is only performed on a single graph. Next, we try to build the assignment matrix from these two association graphs, which are calculated to represent the similarity between the nodes.

$$W_A = \sum_{k=1}^{T} \mathbf{A}_a^\top \cdot Norm_1(\mathbf{A}_a)$$
$$W_D = \sum_{k=1}^{T} \mathbf{A}_d^\top \cdot Norm_1(\mathbf{A}_d) \tag{4}$$

The resulting matrices $W_A$ and $W_D$ provide insights into the similarity structure among the eigenvectors in $F$, delineated by their associations with matrix $F_\tau$ or $M_c$. Specifically, if two vectors in $F$ engage with vectors in $F_\tau$ or $M_c$ in an akin way (i.e., their dot products with vectors in $F_\tau$ or $M_c$ are analogous), then the corresponding elements of these two vectors in the resultant matrix will be amplified, indicating their analogous patterns of action in their interactions with $F_\tau$ or $M_c$.

The diagonal elements of the similarity matrix contain node-to-node similarity information and the off-diagonal elements contain edge-to-edge similarity information. The implication is to simultaneously maximize the first-order similarity as well as the second-order similarity in the matching results. Mathematically, this problem is unable to find a globally optimal solution in polynomial time. Therefore, researchers have proposed the random walk algorithm to solve the problem efficiently and accurately.

We address this matter at both the instance-level and video-level with the Gm-Rw algorithm. In the graph structure of the Gm-Rw algorithm, our nodes consist of two types: single video affinity and intra-class discriminative properties. The Gm-Rw algorithm defines the redistribution of node weights during the random walk as a linear solution process for the assignment matrix and the current action instances.

**Instance-level affinity prototype.** Multiple action instances within a single video should exhibit high affinity. Therefore, the single-frame features of the current instance should have high affinity with some snippets of other action instances, and we should find them and activate them with the aim of suppressing contextual interference at the same time. Therefore, we use the Gm-Rw algorithm to linearly solve for fragments that are affinity with single-frame features in other action instances:

$$(I - \omega_1{}^2 * W_A) \cdot X_A = I \tag{5}$$

Where $X_A$ represents the solution to the linear equation satisfied by the high affinity of the current video sequence with the single-frame features $F_\tau$, and is the prototype of the affinity snippets we wish to obtain inter-guidance within a single video from the current single-frame features to other action snippets. $I$ participates as an identity matrix, which solves a regularized linear system and is used to find a steady state or equilibrium.

**Video-level affinity prototype.** For intra-class videos, action features show greater variability, while discriminative features play a dominant role. Therefore, we need to collect such discriminative single-frame features. We maintain an online memory bank that stores a certain amount of discriminative features $M_c$ for each class. Then, we use the Gm-Rw to linearly solve for fragments that are affinity with single-frame features from other intra-class videos.

$$(I - \omega_2{}^2 * W_D) \cdot X_D = I \tag{6}$$

Likewise, $X_D$ represents the solution to the linear equation satisfied by the more discriminative affinity of the current video sequence with the $M_c$ from the memory bank and is the affinity prototype of the discriminative snippets we wish to propagate within intra-class videos.

**Prototype guidance with EMA.** Using the aforementioned Gm-Rw algorithm, we successfully determine the dual-level affinity prototypes. Then, we shift our focus to devise a strategy to utilize this affinity prototype to guide more optimal activation of CAS. In certain semi-supervised learning scenarios Wang et al. (2022); Sohn et al. (2020); Hu et al. (2021), the Exponential Moving Average (EMA) can incorporate more historical states in the model learning process, which is necessary to generate pseudo-labels to train the network from the current model predictions. In practice, we propose to utilize the EMA for updating original features $F_i$.

Given the instance-level affinity $X_A$ and video-level discriminative $X_D$, we attempt to leverage these affinity prototypes to update and improve the representation of action features. Furthermore, in each training epoch, our EMA operates at a dual-level scale. In detail, we consider both the affinity solution of the current video and the discriminative solution stored in the memory bank as the discriminative affinity obtained during the current iteration. We update the momentum at different ratios for these two solutions as follows, respectively.

$$\begin{aligned} F_i^* &= \omega_1 X_A + (1 - \omega_1).F_i \\ F_i^* &= \omega_2 X_D + (1 - \omega_2).F_i \end{aligned} \tag{7}$$

The dual-level momentum updating occurs simultaneously with $\omega_1$ and $\omega_2$ representing the updating ratios. We observe that the same hyperparameters $\omega_1$ and $\omega_2$ are used in both the random walk process and the EMA update process, and we will analyze the reasons for this in section 3.4.

## 3.4 ADAPTIVE COLLABORATIVE STRATEGY

To achieve dual-level prototypes guidance for higher-quality pseudo-labels of action instance, we summarize instance-level affinity prototype $F_\tau$ and video-level affinity prototype $M_c$. $F_\tau$ is derived from the current video, whose action instances have more similar action representations and show stable and sustainable impact. Thus, we set $\omega_1$ to a constant value to maintain this sustainable guidance. Differently, when video-level affinity prototype guidance is implemented, $M_c$ summarizes more discriminative affinity prototypes under the selection mechanism. To balance the contributions between the instance-level prototype and the video-level prototype, we introduce the adaptive collaborative strategy for the dual-level prototypes guidance as shown in Fig. 2. We set $\omega_2$ to a variable that changes with the training epoch. We compute $\omega_2$ as follows:

$$\omega_2 = \lambda \cdot \tanh(\Delta \cdot (epoch - start)) + \omega_1 \tag{8}$$

Where $\lambda$ and $\Delta$ are hyper-parameters representing the amplitude of the change of the dynamic weights. At the instance-level, the performance of affinity is relatively stable, and specifying a constant $\omega_2$ during model training could maintain optimal performance. As the weight of $\omega_2$ changes dynamically, the classification model first focuses on the current similar parts and gradually starts to focus on the parts with high discriminative affinity snippets of actions. The single focus of the model is alleviated, and comprehensive prototype guidance of the entire instance is further achieved. Comprehensively, the quality of generated pseudo-labels is greatly improved with ACS.

With the help of the ACS strategy, we have improved our Gm-Rw algorithm to achieve dual-level prototypes guidance as shown in Algorithm 1: the Gm-Rw Algorithm for DPG.

---

**Algorithm 1** the Gm-Rw Algorithm for DPG

---

1: **Input:** $F^T$, $F_\tau$, $M_c$
2: Initialize $\omega_1$, $\lambda$, $\Delta$ and $start$ epoch
3: **for** each epoch E **do**
4:     Compute association matrix $A_a$ and $A_d$
5:         $A_a = Softmax(Norm_1(F_\tau * F^T))$
6:         $A_d = Softmax(Norm_1(M_c * F^T))$
7:     Compute similarity matrix $W_A$ and $W_D$
8:         $W_A = \sum_{k=1}^{T} A_a^\top \cdot Norm_1(A_a)$
9:         $W_D = \sum_{k=1}^{T} A_d^\top \cdot Norm_1(A_d)$
10:    Compute $\omega_2$:
11:        $\omega_2 = \lambda \cdot \tanh(\Delta \cdot (E - start)) + \omega_1$
12:    Random Walk on two similarity matrixs:
13:        $(I - \omega_1{}^2 * W_A) \cdot X_A = I$
14:        $(I - \omega_2{}^2 * W_D) \cdot X_D = I$
15:    Dual-level prototypes guidance with EMA:
16:        $F_i^* = \omega_1 * X_A/2 + \omega_2 * X_D/2 + (2 - \omega_1 + \omega_2)/2 * F_i$
17:    return $F^*$
18: **Output:** Refined $F^*$

---

# 4 EXPERIMENTS

## 4.1 DATASETS AND EVALUATION METRICS

**Dataset.** Four datasets: GTEA Fathi et al. (2011) consists of 28 videos of 7 daily actions in the kitchen. They are split into 21 samples for training and 7 samples for testing, with an average of 17.5 single-frame labels per training sample. BEOID Fathi et al. (2011) provides 58 video samples with 30 action classes with an average duration of 60s, with an average of 12.5 action instances per video. THUMOS14 Jiang et al. (2014) has 413 untrimmed videos of 20 action categories, where 200 validation samples are used for training and 213 test samples for performance evaluation. This dataset is challenging due to the varying lengths and diverse occurrence frequencies of action instances. ActivityNet v1.3 Heilbron et al. (2015) consists of 10024 training videos, 4926 validation videos, and 5044 testing videos belonging to 200 action categories.

Table 1: Results on THUMOS14 testing set. We report the mAP values at different IoU thresholds. And I3D denotes the I3D features. ∗ means the methods utilize the additional weak supervision.

| Supervision | Method | mAP(%)@IoU | | | | | | | AVG | | |
| | | 0.1 | 0.2 | 0.3 | 0.4 | 0.5 | 0.6 | 0.7 | (0.1:0.7) | (0.1:0.5) | (0.3:0.7) |
| --- | --- | --- | --- | --- | --- | --- | --- | --- | --- | --- | --- |
| Frame-level (Full) | BSN Lin et al. (2018) | - | - | 53.5 | 45.0 | 36.9 | 28.4 | 20.0 | - | - | 36.8 |
| | BMN Lin et al. (2019) | - | - | 56.0 | 47.4 | 38.8 | 29.7 | 20.5 | - | - | 38.5 |
| | BSN++ Su et al. (2021) | - | - | 59.9 | 49.5 | 41.3 | 31.9 | 22.8 | - | - | 41.1 |
| Video-level (Weak) | DCC(I3D) Li et al. (2022a) | 69.0 | 63.8 | 55.9 | 45.9 | 35.7 | 24.3 | 13.7 | 44.0 | 54.1 | 35.1 |
| | ASM-Loc(I3D) He et al. (2022) | 71.2 | 65.5 | 57.1 | 46.8 | 36.6 | 25.2 | 13.4 | 45.1 | 55.4 | 35.8 |
| | TEN(I3D) Li et al. (2022b) | 69.7 | 64.5 | 58.1 | 49.9 | 39.6 | 27.3 | 14.2 | 46.1 | 56.3 | 37.8 |
| | RSKP(I3D) Huang et al. (2022) | 71.3 | 65.3 | 55.8 | 47.5 | 38.2 | 25.4 | 12.5 | 45.1 | 55.6 | 35.9 |
| | DELU(I3D) Chen et al. (2022) | 71.5 | 66.2 | 56.5 | 47.7 | 40.5 | 27.2 | 15.3 | 46.4 | 56.5 | 37.4 |
| | P-MIL(I3D) Ren et al. (2023) | 71.8 | 67.5 | 58.9 | 49.0 | 40.0 | 27.1 | 15.1 | 47.0 | 57.4 | 38.0 |
| | RFBS(I3D) Liu et al. (2023) | 72.3 | - | 59.2 | - | 37.7 | - | 13.7 | 46.4 | 57.1 | - |
| | DDG-Net(I3D) Tang et al. (2023) | 75.1 | 68.9 | 60.2 | 48.9 | 38.3 | 26.8 | 14.7 | 47.2 | 58.2 | 37.7 |
| | PivoTAL(I3D) Rizve et al. (2023) | 74.1 | 69.6 | 61.7 | 52.1 | 40.0 | 27.1 | 15.1 | 47.0 | 57.4 | 38.0 |
| Frame-level (Weak∗) | STAR(I3D) Xu et al. (2019) | 68.8 | 60.0 | 48.7 | 34.7 | 23.0 | - | - | - | 47.0 | - |
| | SF-Net(I3D) Ma et al. (2020) | 68.3 | 62.3 | 52.8 | 42.2 | 30.5 | 20.6 | 12.0 | 41.2 | 51.2 | 31.6 |
| | DCM(I3D) Ju et al. (2021) | 72.8 | 64.9 | 58.1 | 46.4 | 34.5 | 21.8 | 11.9 | 44.3 | 55.3 | 34.5 |
| | LAC(I3D) Lee & Byun (2021) | 75.7 | 71.4 | 64.6 | 56.5 | 45.3 | 34.5 | 21.7 | 52.8 | 62.7 | 45.8 |
| | ARIM(I3D) Yu et al. (2023) | 73.1 | 66.8 | 58.6 | 47.9 | 37.0 | 24.3 | 12.8 | 45.8 | 56.6 | 36.1 |
| | CRRC-Net(I3D) Fu et al. (2022) | 77.8 | 73.5 | 67.1 | 57.9 | 46.6 | 33.7 | 19.8 | 53.8 | 64.6 | 45.1 |
| | NGPR(I3D) Li et al. (2024) | 77.9 | 73.9 | 66.6 | 59.4 | 48.6 | 36.7 | 22.7 | 55.1 | 65.3 | 46.8 |
| | Ours(I3D) | **81.8** | **77.0** | **70.3** | **61.2** | **51.2** | **37.2** | **23.1** | **57.4** | **67.0** | **47.5** |

Table 2: Results on GTEA and BEOID validation sets. AVG means the average mAP from IoU=0.1 to 0.7.

| Method | GTEA | | | | | BEOID | | | | |
| | mAP(%)@IoU | | | | | mAP(%)@IoU | | | | |
| | 0.1 | 0.3 | 0.5 | 0.7 | AVG | 0.1 | 0.3 | 0.5 | 0.7 | AVG |
| --- | --- | --- | --- | --- | --- | --- | --- | --- | --- | --- |
| SF-Net Ma et al. (2020) | 58.0 | 37.9 | 19.3 | 11.9 | 31.0 | 62.9 | 40.6 | 16.7 | 3.5 | 30.9 |
| DCM Ju et al. (2021) | 59.7 | 38.3 | 21.9 | 18.1 | 33.7 | 63.2 | 46.8 | 20.9 | 5.8 | 34.9 |
| Li et al. Li et al. (2021) | 60.2 | 44.7 | 28.8 | 12.2 | 36.4 | 71.5 | 40.3 | 20.3 | 5.5 | 34.4 |
| LAC Lee & Byun (2021) | 63.9 | 55.7 | 33.9 | 20.8 | 43.5 | 76.9 | 61.4 | 42.7 | 25.1 | 51.8 |
| NGPR Li et al. (2024) | 74.3 | 62.8 | 35.7 | 13.7 | 46.6 | 77.2 | 64.3 | 44.0 | 24.5 | 53.1 |
| Ours(I3D) | **71.8** | **61.5** | **37.8** | **16.7** | **46.9** | **77.3** | **69.7** | **53.7** | **25.5** | **56.5** |

**Evaluation Metrics.** We follow the standard evaluation protocol by reporting mean average precision (mAP) values under different intersection over union (IoU) thresholds.

## 4.2 IMPLEMENTATION DETAILS

We employ the Inflated 3D ConvNet (I3D) Gkalelis et al. (2009), pre-trained on the Kinetics-400 dataset, as our feature extractor. The embedded module consists of a 1D conventional layer with the rectified linear unit (ReLU) activation function. Each video is segmented into 16-frame segments, which serve as inputs to the feature extractor, yielding 2048-dimensional late-fusion features. We maintain the original number of segments as $\psi$ without any form of sampling. Our model is optimized using the Adam optimizer, with a learning rate set to 1e-4 and a batch size of 16. Hyperparameters are selected via grid search, with $\gamma = 0.95$ and $\psi = 0.1$. The video-level threshold $\theta^{vid}$ is established at 0.5, while the segment-level threshold $\theta^{seg}$ ranges from 0 to 0.25, with increments of 0.05. Non-maximum suppression (NMS) is applied with a threshold of 0.7. In adaptive collaborative strategy, we set $\omega_1 = 0.2$, $\omega_2 = 0.04$, $\lambda = 0.2$ and $\Delta = 0.0025$ for dynamically update to balance dual-level prototypes guidance.

## 4.3 COMPARISON WITH STATE-OF-THE-ART METHODS

We report experimental results for DPG in Table 1, Table 2, and Table 3. These results are based on the THUMOS14, BEOID, GTEA, and ActivityNet1.3 validation datasets. The proposed method outperforms state-of-the-art methods on the datasets. First, our model significantly outperforms state-of-the-art point-level weakly supervised action localization methods. On THUMOS14, our

Table 3: Results on ActivityNet1.3 validation set. AVG means the average mAP from IoU=0.5 to 0.95.

| ActivityNet1.3 | mAP(%)@IoU | | | |
|---|---|---|---|---|
| | 0.5 | 0.75 | 0.95 | AVG |
| LAC Lee & Byun (2021) | 40.4 | 24.6 | 5.7 | 25.1 |
| CRRC Fu et al. (2022) | 39.8 | 24.1 | 5.9 | 24.0 |
| NGPR Li et al. (2024) | 41.3 | 30.9 | 4.8 | 26.5 |
| Ours(I3D) | **42.4** | **31.6** | **6.3** | **26.7** |

Table 4: Ablation studies on the THUMOS14. The model with $\mathcal{L}_{base}$ is our baseline. And the model with instance-level and video-level prototype guidance is the full model.

| $\mathcal{L}_{base}$ | $ins-level$ | $vid-level$ | mAP(%)@IoU | | | | |
|---|---|---|---|---|---|---|---|
| | | | 0.1 | 0.3 | 0.5 | 0.7 | AVG |
| ✓ | - | - | 75.7 | 64.6 | 45.3 | 21.8 | 52.8 |
| ✓ | ✓ | - | 78.6 | 67.3 | 45.8 | 22.8 | 54.5 |
| ✓ | - | ✓ | 79.9 | 69.2 | 49.1 | 22.7 | 56.3 |
| ✓ | ✓ | ✓ | **81.3** | **70.3** | **51.2** | **23.1** | **57.4** |

DPG significantly improves mAP by 2.3% over NGPR at AVG mAP (0.1:0.7). Under low IoU thresholds, our performance advantage is great, for example, the improvement of mAP@IoU = 0.1 is 3.9%, and the improvement of mAP@IoU = 0.2 is 3.1%. It is worth noting that our point-supervised method can surpass the performance of fully supervised methods such as BSN++ (6.4% higher at AVG mAP (0.3:0.7)). In addition, in terms of AVG mAP (0.1:0.7), compared with NGPR, our method significantly improved by 0.3% on GTEA, 3.4% on BEOID, and 0.2% on ActivityNet1.3. The effectiveness of the proposed method and its versatility on different datasets are demonstrated.

### 4.4 ABLATION STUDY

**Effectiveness of dual-level knowledge guidance.** As shown in Table 4, we study the two major elements of dual-level guidance, namely instance-level guidance for improving single-frame discrimination and video-level guidance for improving single-frame completeness. The mAP@IoU from 0.1 to 0.7 of base-line are 75.7%, 64.6%, 45.3%, and 21.8% respectively. After the baseline is guided by instance-level prototype, we observe that the above performance indicators are improved, which highlights the importance of instance-level guidance and verifies that our method is also effective in the case of a single video. In addition, after introducing video-level guidance into the baseline, our performance indicators are improved by 4.2%, 4.6%, 3.8%, and 0.9%, respectively, with an AVG mAP improvement of 3.5%. Finally, by integrating the two guidance techniques, the proposed method achieves the best performance in all five indicators compared with the baseline. For example, the mAP@IoU from 0.1 to 0.7 is 5.6%, 5.7%, 5.9%, and 1.3% higher than the baseline, and the AVG mAP is improved by 4.6%. This shows that the proposed instance-level prototype and video-level prototype guidance can work together and prove the respective effect of the dual-level prototypes guidance.

**Effectiveness of $ACS$.** In the above, we proposed a collaborative optimization strategy to improve the effect of instance-level prototype guidance and video-level prototype guidance on the entire model. As shown in Table 5, the full model with $ACS$ has improved in every benchmark indicator compared to the full model without $ACS$, and the AVG has increased by 1.1%, which is very consistent with the purpose of designing $ACS$, that is, to ensure that the prototype guidance in DPG is diverse within the same class and encourage extensive learning of different features of actions within the class. This also means that the collaborative optimization method of multiple instances and multiple videos also promotes the improvement of model performance.

**Effectiveness of $M$.** Above, we proposed using memory banks to store features of the same class to improve the integrity of single-frame information. Table 6 reports experiments with the number of features $M$ in the memory bank from 2 to 11. We can conclude that the value of $M$ has a significant impact on the performance of our model and $M = 5$ can obtain the best result 57.5%. We found that when $M$ is set too small, the learned features are not complete enough and cannot function as a memory bank. When $M$ is set too large, there is redundant single-frame information in a single frame in the memory bank, and the effect is not significantly improved.

Table 5: Results on THUMOS14 testing set. AVG means the average mAP from IoU=0.1 to 0.7.

| Method | mAP(%)@IoU | | | | |
|---|---|---|---|---|---|
| | 0.1 | 0.3 | 0.5 | 0.7 | AVG |
| without $ACS$ | 80.1 | 68.8 | 48.9 | 22.1 | 56.3 |
| with $ACS$ | **81.3** | **70.3** | **51.2** | **23.1** | **57.4** |

Table 6: Results of different $M$ values on video-level and dual-level.

| Method | AVG mAP(%)(0.1:0.7) | | | | | | |
|---|---|---|---|---|---|---|---|
| | 2 | 3 | 4 | 5 | 7 | 9 | 11 |
| video-level | 55.4 | 55.9 | 56.1 | 56.3 | 56 | 56.1 | 56.1 |
| dual-level | **55.9** | **56.7** | **57.3** | **57.5** | **57.5** | **57.4** | **57.4** |

## 4.5 QUALITATIVE RESULTS

To demonstrate the effectiveness of the proposed method, we visualize several examples of localized action instances in Fig. 3. These results are from the THUMOS14 testing set, where Baseline, Ours, and GT stand for results on LAC, our method, and the ground truth, respectively. Three examples are selected: CliffDiving, FrisbeeCatch1 and FrisbeeCatch2. First, when the three videos are compared with the baseline, the proposed method DPG improves the completeness of the localization performance. Second, it can be observed in CliffDiving that the co-occurring background, as well as highly similar affinity foreground actions, exist between multiple instances of the same video. Third, there are associated clips between different videos of the same class, as shown in FrisbeeCatch1 and FrisbeeCatch2, which are guided by the modules we designed so that they can interact with each other and guide to promote pseudo-labels.

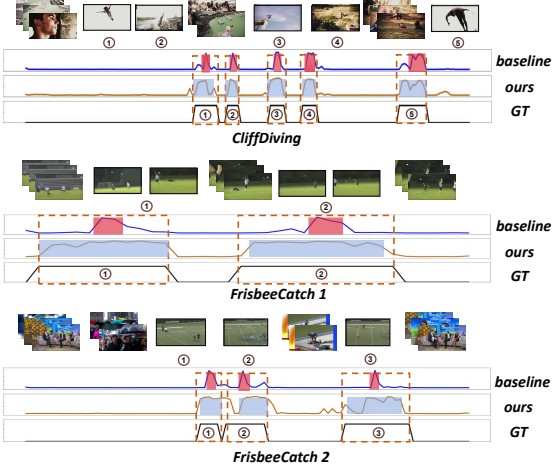

Figure 3: Comparison of effectiveness. We show the two class three untrimmed videos, namely, CliffDiving, FishbeeCatch1, and FishbeeCatch2. GT is the ground truth of the videos, the baseline is the LAC Lee & Byun (2021), and DPG is the method we proposed.

## 5 CONCLUSION

We expand upon previous STAL methods which focused on generating pseudo-labels to catch up with the accuracy of fully-supervised. Unlike previous pseudo-labels methods, we extend the scope of generating pseudo-labels from corresponding instances of single-frame to dual-level intra-class instances. Thus, we refine pseudo-labels from the perspective of instance-level and video-level affinity prototype. Innovatively, we design the Gm-Rw algorithm to explore the affinity prototype, and then achieve the collaborative dual-level prototypes guidance under the EMA mechanism. The evaluation of our method on three benchmark datasets, including THUMOS14, GTEA, BEOID, and ActivityNet1.3 demonstrates that our proposed method achieves state-of-the-art performance.

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
