# A APPENDIX

## A.1 RELATED WORK

**Fully-Supervised Temporal Action Localization** Fully-supervised TAL like Two-Stream CNNs Ye et al. (2015) and I3D Gkalelis et al. (2009) have excelled, utilizing labeled action annotations for precise localization, can be divided into two groups. The first type employs a two-stage framework Xu et al. (2017); Chao et al. (2018); Shou et al. (2016); Zhu et al. (2021), namely, proposal-plus classification. The second approach designs an end-to-end pipeline by integrating proposal generation and classification. Shou et al. (2016) adopts multi-scale slide windows for the former types to generate proposals and feeds them into a three-segment structure. Other works obtain proposals by predicting each snippet's action probability. SSN Zhao et al. (2017) devices a temporal actionness grouping algorithm and BSN Lin et al. (2018) predicts starting and ending probabilities of each snippet to generate proposals. Furthermore, the later paradigm is proposed to relieve the computation burden on the two-stage frameworks. For instance, BMN Lin et al. (2019) proposes a boundary-matching unit for evaluating the confidence of densely distributed proposals. However, weakly supervised TAL methods have gained prominence for their ability to localize actions with only video-level labels, making them more adaptable for real-world scenarios.

**Weakly-supervised Temporal Action Localization** Without precise start and end timestamps as labels, existing WTAL frameworks are mainly divided into three types. The first type generates video-level features by attention sequence Nguyen et al. (2018); Islam et al. (2021) and utilizes a classifier to predict the video class. The second model produces CAS directly and adopts top-k operation to obtain the video-level scores Yuan et al. (2019). The others get the video-level scores via combining CAS and attention sequence Islam et al. (2021), which is also adapted in our work. However, class-specific background snippets will be activated while hard action snippets will be ignored for the classify loss. To overcome the problems, Kumar Singh & Jae Lee (2017); Zhang et al. (2019); Zhong et al. (2018); Zeng et al. (2019) present an erasing strategy to mine the less discriminative action snippets. Differently, some methods Lee et al. (2020); Liu et al. (2021a) annotate the background snippets as an individual class to suppress the class-specific background snippets. Meanwhile, many works also deal with this issue in different ways. One popular strategy Luo et al. (2020); Liu et al. (2019) is constructing frame-wise pseudo labels. For example, ACSNet Liu et al. (2021b) generates positive/negative snippets by exploiting the differences between RGB and optical flow modalities. Moreover, the temporal relationship of actions also has drawn much attention. Shou et al. (2016); Zhao et al. (2017); Yu et al. (2019); Jenni & Jin (2021) generate action/background proposals by pseudo labels and introduce the specific constraints to separate them. ASM-Loc He et al. (2022) performs action-aware segment modeling with action proposals for capturing the temporal structures to suppress noisy backgrounds. DCC Li et al. (2022a) denoises the pseudo labels to obtain robust features and design a diverse contrast learning strategy to enable action-background separation. RSKP Huang et al. (2022) refines pseudo labels via representative snippet knowledge propagation, whose pseudo labels are generated from the temporal class activation maps of to rectify the predictions of the classification model.

**Single-frame Temporal Action Localization.** To balance labeling cost and model performance, single-frame temporal action localization (STAL) task has been proposed. Bearman et al. Bearman et al. (2016) first utilized the single-point supervision for image semantic segmentation. Mettes et al. Mettes et al. (2016) extended it to spatio-temporal localization in video, where the action is pointed out by one spatial location in each action frame. The success in these fields demonstrates that single-frame supervision can achieve very promising results already. Recently, single-frame supervision has been used in for temporal action classification and localization tasks, which bridges the gap between weakly supervised methods and full supervised methods. Under single-frame setting, the models are assisted by training by marking the timestamp and action category of one frame for each action instance in each video, has addressed identifying temporal boundaries. Note that Alwassel et al. Alwassel et al. (2018) proposed to spot action in the video during inference time but targeted detecting one action instance per class in one video while our proposed single-frame localization task aims to detect every instancbe in one video. Sf-net Ma et al. (2020) was the first work that began to use frame-level labels. Chen et al. Ju et al. (2021) propose a method of performing boundary regression using point-supervised key frame prediction. Then, some studies Ju et al. (2021); Lee & Byun (2021); Li et al. (2024) proposed a pseudo-labels mining strategy, which is based on single frame annotations to obtain more video frame information by mining more pseudo-action frames

and pseudo-background frames. Based on focus loss, works such as Lee & Byun (2021) propose to generate a dense optimal sequence method sequence and use an attention mechanism to separate action instance and background. And NGPR Li et al. (2024) uses neighbor features to enhance the representation of action snippets for pseudo labels refinement.

## A.2 SUPPLEMENT OF BASELINE

We adopt extracted features as our input. The features include RGB features $F^{rgb} \in \mathbb{R}^{T \times D}$ and optical flow features $F^{flow} \in \mathbb{R}^{T \times D}$, where $T$ and $D$ are the number of snippets and the dimension, respectively. We only use $F$ to denote the two kinds of features. In addition, we obtain the corresponding single-frame features based on the single-frame annotations $\tau$. The video-level class label $p_i$ can be derived by accumulating the point-level labels using the temporal top-k pooling for aggregation. The video-level classification loss is calculated with binary cross-entropy.

$$\mathcal{L}_{vid} = -\sum_{i=1}^{C}(y_i log p_i + (1 - y_i)log(1 - p_i))) \tag{9}$$

The point-level classification loss is also computed by binary cross-entropy but includes the background for effective training. Formally, the point-level classification loss is defined as follows.

$$\mathcal{L}_{point}^{act} = \sum_{c=1}^{C}\left(y_i\left(1 - \hat{p}_i[\tau]\right)^{\beta}\log \hat{p}_i[\tau] + (1 - y_i[\tau])\,p_i[\tau]^{\beta}\log\left(1 - \hat{p}i[\tau]\right)\right) \tag{10}$$

Likewise, we obtain $\mathcal{L}_{point}^{bkg}$ for background points. With this, our point-level classification loss $\mathcal{L}_{point}$ can be expressed as follows:

$$\mathcal{L}_{point} = \mathcal{L}_{point}^{act} + \alpha * \mathcal{L}_{point}^{bkg} \tag{11}$$

Where $\alpha$ represents a hyper-parameter. we consider the video-level classification loss, $\mathcal{L}_{vid}$, along with the point-level classification loss, $\mathcal{L}_{point}$, as the foundational elements of our method. By leveraging feature and score contrastive learning, we are able to realize the completeness of learning within the current videos.

To this end, we introduce $\mathcal{L}_{comp}$ to facilitate completeness learning by contrasting action instances with background instances. Furthermore, drawing inspiration from the recent advancements in contrastive learning, we devise a feature contrastive loss along the temporal dimension, where $f^{\pm}$ denotes positive and negative samples. Consequently, our comprehensive completeness learning framework comprises two essential components.

$$\mathcal{L}_{comp} = \frac{1}{\sum_{i=1}^{C} y_i}\sum_{i=1}^{C} y_i(1 - score[i])^{\sigma} - \frac{1}{\sum_{n=1}^{N} \pi_n^c}\sum_{n=1}^{N} \pi_n^c log\frac{\sum_{f^+ \in F} exp(f^{\tau}.f^+/\varphi)}{\sum_{f^{\pm} \in F} exp(f^{\tau}.f^{\pm}/\varphi)} \tag{12}$$

Where $\sigma$ is a hyper-parameter. Finally, we get the baseline:

$$\mathcal{L}_{base} = \mathcal{L}_{vid} + \mathcal{L}_{point} + \mathcal{L}_{comp} \tag{13}$$

The combination of these three components forms the foundational baseline of our DPG method, which serves as a starting point for further refinement and enhancement.