# OpenReview forum: "Dual-level Prototypes Guidance for Single-frame Temporal Action Localization"
_ICLR.cc/2025/Conference — ICLR 2025 Conference Withdrawn Submission_

### Official Review · Reviewer_PDmZ · 2024-10-31

**Soundness:** 1
**Presentation:** 2
**Contribution:** 2
**Rating:** 5
**Confidence:** 5

**Summary:**

The paper proposes a dual-level prototype guidance method for pseudo-labels refinement in single-frame temporal action localization (STAL). For instance-level guidance, the method leverages the affinity between single-frame features and action instances to enhance action representations. For video-level guidance, the method stores more discriminative prototypes in a memory bank to provide intra-class prototype guidance. The proposed method consistently outperforms state-of-the-art STAL methods across multiple datasets.

**Strengths:**

1.The proposed method refines pseudo-labels for STAL from a prototype view. This is a good direction.
2.Experiment results on multiple datasets demonstrate that our proposed method achieves state-of-the-art results.
3. The paper provides a large body of ablation studies for each involved design component individually.

**Weaknesses:**

1. The symbols in the formula have not been fully defined, such as sort() in Eq. (1) and $M_{c}$ in Eq. (2). Please further explain them.
2. There is a lack of ablation experiments regarding the hyperparameter w1 in Eq. (5).
3. I wonder if there are similar methods to replace GM algorithm for solving equations 5 and 6. Better provide ablation experiments regarding this.
4. The writing needs further polishing. For example, Line 84 “and diversity supports the feature completeness. ” -> “ensuring that diversity supports the completeness of the features”
5. More comparison results regarding the training and inference costs and speeds need to be provided.

**Questions:**

1. The symbols in the formula have not been fully defined, such as sort() in Eq. (1) and $M_{c}$ in Eq. (2). Please further explain them.
2. There is a lack of ablation experiments regarding the hyperparameter w1 in Eq. (5).
3. I wonder if there are similar methods to replace GM algorithm for solving equations 5 and 6. Better provide ablation experiments regarding this.
4. The writing needs further polishing. For example, Line 84 “and diversity supports the feature completeness. ” -> “ensuring that diversity supports the completeness of the features”
5. More comparison results regarding the training and inference costs and speeds need to be provided.

---

### Official Review · Reviewer_HgDt · 2024-10-31

**Soundness:** 1
**Presentation:** 1
**Contribution:** 2
**Rating:** 3
**Confidence:** 5

**Summary:**

Overall, the paper introduces an intriguing method for single-frame temporal action localization by presenting a novel dual-level prototype guidance architecture. This architecture effectively tackles the critical challenges of insufficient feature discrimination and incomplete feature representation in single-frame temporal action localization.  However, the paper's technical contributions are not sufficiently novel, as well as lacking important references. Furthermore, the writing of the paper is poor and could be improved for clarity and coherence.

**Strengths:**

1. The paper proposes a novel dual-level prototype guidance architecture that addresses the challenges of insufficient feature discrimination and lacking feature completeness in single-frame temporal action localization.
2. The proposed method has the potential to advance the field of temporal action localization, particularly in scenarios with limited supervision

**Weaknesses:**

1. The proposed method of this paper is not very novel, and the contributions are limited. Specifically, the proposed Dual-Level Prototypes Guidance (DPG) method in this paper, while addressing the challenge of single-frame temporal action localization (STAL) with sparse annotations, may seem to build upon existing pseudo-label strategies. However, its innovative use of the graph matching random walk (Gm-Rw) algorithm for instance-level and video-level prototype guidance distinguishes it from prior work. Overall, the novelty of this work is far from reaching the level of publication.
2. This paper, while presenting a comprehensive analysis of STAL methods, may have omitted many key references such as "HR-PRO," among others. To ensure a thorough comparison, the authors should include these references and update Tables 1, 2, and 3 to incorporate the latest state-of-the-art methods, including:
 [1] HR-Pro: Point-supervised Temporal Action Localization via Hierarchical Reliability Propagation (AAAI24)
 [2] Enhancing single-frame supervision for better temporal action localization. TVCG 24
 [3] Stepwise Multi-grained Boundary Detector for Point-Supervised Temporal Action Localization. ECCV
 .....
3. The introduction of this paper may have lacked clarity in summarizing its contributions. The summary provided by the authors from L105 is uninformative, and thus the authors need to emphasize specific contributions and innovations.
4. Many symbols in the paper are strange. For example, may be \cdot in Eq. (7), \times in Eq. (3), no explanation about W_D in Eq(6), W_A in Eq(5).  L231 'Where'? may be 'where'.  A syntax error occurred in the description of figure 3.   ... and many many....
5. The presentation of this paper is unclear and challenging to comprehend. The writing lacks professionalism and requires substantial enhancement for better readability and coherence. Even more, the abstract of this paper is also unprofessional.
6. The citation format in the references section is inconsistent, containing numerous inaccuracies that need to be addressed for compliance with academic standards. For example, Tan Yu, et al, C. Zhang et al, P. Zhang, and many lack pages, and the wrong format such as  Zikang Yu et al..... I ensure very few formats are correct.

**Questions:**

1. The proposed method of this paper is not very novel, and the contributions are limited. Specifically, the proposed Dual-Level Prototypes Guidance (DPG) method in this paper, while addressing the challenge of single-frame temporal action localization (STAL) with sparse annotations, may seem to build upon existing pseudo-label strategies. However, its innovative use of the graph matching random walk (Gm-Rw) algorithm for instance-level and video-level prototype guidance distinguishes it from prior work. Overall, the novelty of this work is far from reaching the level of publication.
2. This paper, while presenting a comprehensive analysis of STAL methods, may have omitted many key references such as "HR-PRO," among others. To ensure a thorough comparison, the authors should include these references and update Tables 1, 2, and 3 to incorporate the latest state-of-the-art methods, including:
 [1] HR-Pro: Point-supervised Temporal Action Localization via Hierarchical Reliability Propagation (AAAI24)
 [2] Enhancing single-frame supervision for better temporal action localization. TVCG 24
 [3] Stepwise Multi-grained Boundary Detector for Point-Supervised Temporal Action Localization. ECCV
 .....
3. The introduction of this paper may have lacked clarity in summarizing its contributions. The summary provided by the authors from L105 is uninformative, and thus the authors need to emphasize specific contributions and innovations.
4. Many symbols in the paper are strange. For example, may be \cdot in Eq. (7), \times in Eq. (3), no explanation about W_D in Eq(6), W_A in Eq(5).  L231 'Where'? may be 'where'.  A syntax error occurred in the description of figure 3.   ... and many many....
5. The presentation of this paper is unclear and challenging to comprehend. The writing lacks professionalism and requires substantial enhancement for better readability and coherence. Even more, the abstract of this paper is also unprofessional.
6. The citation format in the references section is inconsistent, containing numerous inaccuracies that need to be addressed for compliance with academic standards. For example, Tan Yu, et al, C. Zhang et al, P. Zhang, and many lack pages, and the wrong format such as  Zikang Yu et al..... I ensure very few formats are correct.

---

### Official Review · Reviewer_qAEL · 2024-11-03

**Soundness:** 2
**Presentation:** 2
**Contribution:** 1
**Rating:** 3
**Confidence:** 4

**Summary:**

This paper addresses the challenges of single-frame temporal action localization, particularly the issue of sparse single-frame annotations. The authors propose a dual-level prototypes guidance method combined with the graph matching random walk algorithm to refine pseudo-labels by leveraging intra-class affinity. The method introduces an adaptive collaborative strategy for dynamic optimization, which enhances the refinement process. Extensive experiments on major datasets demonstrate the effectiveness.

**Strengths:**

A. The paper introduces a dual-level prototype guidance (DPG) approach, using both instance-level and video-level prototypes to improve pseudo-labels for action localization.
B. Baseline comparison is thorough.
C. The paper is written in clear, making it easy to comprehend.

**Weaknesses:**

A. The paper lacks sufficient discussion on the issues caused by sparse annotations, which reduces its readability. It should elaborate on why sparse annotations lead to these issues.
B. In line 102, it is unclear why dual-level prototype guidance can achieve a generalized vision framework and whether there is experimental evidence to support this claim. Additionally, it should be clarified why this feature is referred to as "Flexibility."
C. The meaning of the equations is unclear, specifically whether A in Equation (4) is the same as A in Equation (3).
D. The paper does not address the potential computational cost of the proposed method. It should provide a comparison of the computational costs, such as runtime, parameter count, and memory consumption.
E. The paper lacks ablation experiments for certain hyperparameters. It should clarify how parameters like \lambda and \omega_1 in Equation (8) impact \omega_2  and ultimately affect model performance.
F. Equations lack necessary punctuation.

**Questions:**

please see the above weaknesses.

---

### Official Review · Reviewer_2mEU · 2024-11-03

**Soundness:** 4
**Presentation:** 3
**Contribution:** 2
**Rating:** 5
**Confidence:** 4

**Summary:**

The paper tackles the single-frame temporal action detection task. It refines the pseudo labels by introducing a dual-level prototype guidance approach to highlight intra-class affinity and discrimination from within videos and intra-class videos for better pseudo guidance, and an adaptive collaborative strategy to facilitate dynamic optimization in training. Extensive experiments show improved metrics on classic datasets.

**Strengths:**

- The dual-level prototype guidance is new and proven effective by ablations on THUMOS14.
- Qualitative results show superior mAP@IoU results on classic TAD datasets.

**Weaknesses:**

- Missing Comparison. Some advanced full TAD approaches are missing in Table 1 and 3, for example, Actionformer[M1], TadTR[M2], etc.
- Backbone. I3D is one of the traditional fully convolution-based video backbones compared to the more advanced Omnivore[M3] and VideoMAE[M4]. It would be nice to see if the proposed design works on more advanced backbone features.
- The mAP @ IoU = 0.1 and 0.3 results on GTEA in Table 2 are marginally worse than NGPR (please also fix the bold annotations for these entries). Also, we observe that the mAP improvement is large on smaller datasets (THUMOS14 and BEOID), but the improvement on the large-scale dataset, i.e. ActivityNet, is small. The authors should provide more insights and analysis concerning the above observations in the experiment section.
- Learning the prototypes is shown to work well under pre-defined video categories and single-label videos, but its potential to extend to more complex scenarios: open-vocabulary, multi-label videos are questionable, which could reduce its impact in the community and downstream applications.
- [Minor, Writing issues] Please check for spelling and grammar errors. Write out the full name of an abbreviation the first time it gets mentioned, i.e. DBP is directly used without explaining the full name.

[M1] Zhang, Chen-Lin, Jianxin Wu, and Yin Li. "Actionformer: Localizing moments of actions with transformers." ECCV, 2022.

[M2] Liu, Xiaolong, et al. "End-to-end temporal action detection with transformer." TPAMI.

[M3] Girdhar, Rohit, et al. "Omnivore: A single model for many visual modalities." CVPR, 2022.

[M4] Wang, Limin, et al. "Videomae v2: Scaling video masked autoencoders with dual masking." CVPR, 2023.

**Questions:**

Please refer to the weakness section.

---

### Note · Authors · 2024-11-26

**Comment:**

We would like to express our sincere gratitude for the valuable feedback and suggestions provided on our manuscript. After carefully considering the reviewers' comments, we recognize that there are areas in our paper that require further revision and enhancement to achieve a higher quality standard. To address these concerns thoroughly, we have decided to withdraw our current submission.
Additionally, we believe that this decision will help reduce the workload of the Area Chair by avoiding repeated reviews. We deeply appreciate the guidance and support from the editorial team and reviewers.

**Withdrawal Confirmation:**

I have read and agree with the venue's withdrawal policy on behalf of myself and my co-authors.